# Radiotherapy for Recurrent Medulloblastoma in Children and Adolescents: Survival after Re-Irradiation and First-Time Irradiation

**DOI:** 10.3390/cancers16111955

**Published:** 2024-05-22

**Authors:** Jonas E. Adolph, Gudrun Fleischhack, Sebastian Tschirner, Lydia Rink, Christine Dittes, Ruth Mikasch, Philipp Dammann, Martin Mynarek, Denise Obrecht-Sturm, Stefan Rutkowski, Brigitte Bison, Monika Warmuth-Metz, Torsten Pietsch, Stefan M. Pfister, Kristian W. Pajtler, Till Milde, Rolf-Dieter Kortmann, Stefan Dietzsch, Beate Timmermann, Stephan Tippelt

**Affiliations:** 1Department of Pediatrics III, Center for Translational Neuro- and Behavioral Sciences (CTNBS), University Hospital of Essen, 45122 Essen, Germany; gudrun.fleischhack@uk-essen.de (G.F.); sebastian.tschirner@uk-essen.de (S.T.); lydia.rink@uk-essen.de (L.R.); christine.gaab.1@stud.uni-due.de (C.D.); stephan.tippelt@uk-essen.de (S.T.); 2Department of Neurosurgery and Spine Surgery, University Hospital Essen, 45122 Essen, Germany; philipp.dammann@uk-essen.de; 3Department of Pediatric Hematology and Oncology, Center for Obstetrics and Pediatrics, University Medical Center Hamburg-Eppendorf, 20251 Hamburg, Germany; m.mynarek@uke.de (M.M.); d.obrecht-sturm@uke.de (D.O.-S.); s.rutkowski@uke.de (S.R.); 4Mildred Scheel Cancer Career Center HaTriCS4, University Medical Center Hamburg-Eppendorf, 20251 Hamburg, Germany; 5Diagnostic and Interventional Neuroradiology, Faculty of Medicine, University of Augsburg, 86156 Augsburg, Germany; brigitte.bison@uk-augsburg.de; 6Institute of Diagnostic and Interventional Neuroradiology, University Hospital Wuerzburg, 97080 Wuerzburg, Germany; warmuth_m@ukw.de; 7Institute of Neuropathology, DGNN Brain Tumor Reference Center, University Hospital of Bonn, 53105 Bonn, Germany; torsten.pietsch@ukbonn.de; 8Division of Pediatric Neurooncology, German Cancer Research Center (DKFZ), 69120 Heidelberg, Germany; s.pfister@kitz-heidelberg.de (S.M.P.); kristian.pajtler@med.uni-heidelberg.de (K.W.P.); 9Department of Pediatric Oncology and Hematology, University Hospital Heidelberg, 69120 Heidelberg, Germany; till.milde@med.uni-heidelberg.de; 10Hopp Children’s Cancer Center Heidelberg (KiTZ), 69120 Heidelberg, Germany; 11National Center for Tumor Diseases (NCT), 69120 Heidelberg, Germany; 12Clinical Cooperation Unit (CCU) Pediatric Oncology, German Cancer Research Center (DKFZ), 69120 Heidelberg, Germany; 13German Consortium for Translational Cancer Research (DKTK), 69120 Heidelberg, Germany; 14Department of Radio-Oncology, University Leipzig, 04129 Leipzig, Germany; rolf-dieter.kortmann@medizin.uni-leipzig.de (R.-D.K.); stefan.dietzsch@medizin.uni-leipzig.de (S.D.); 15Department of Particle Therapy, University Hospital Essen, West German Proton Therapy Centre Essen, 45122 Essen, Germany; beate.timmermann@uk-essen.de

**Keywords:** medulloblastoma, recurrence, radiotherapy, re-irradiation, resection

## Abstract

**Simple Summary:**

The treatment options for children with recurrent medulloblastoma are often limited due to extensive previously received treatments after initial diagnosis. Especially, repeated radiotherapy is associated with significant side-effects. In this study, we study the impact on survival of repeated radiotherapy at recurrence for patients with previous irradiation, as well as first irradiation at recurrence when no previous radiotherapy was applied. We find that repeated radiotherapy provides a short-time benefit in terms of survival, but survival ten years after recurrence is not significantly improved. At the same time, we find that applying radiotherapy at recurrence when patients received no previous irradiation did significantly improve survival, both short and long term.

**Abstract:**

Background: Radiotherapy (RT) involving craniospinal irradiation (CSI) is important in the initial treatment of medulloblastoma. At recurrence, the re-irradiation options are limited and associated with severe side-effects. Methods: For pre-irradiated patients, patients with re-irradiation (RT2) were matched by sex, histology, time to recurrence, disease status and treatment at recurrence to patients without RT2. Results: A total of 42 pre-irradiated patients with RT2 were matched to 42 pre-irradiated controls without RT2. RT2 improved the median PFS [21.0 (CI: 15.7–28.7) vs. 12.0 (CI: 8.1–21.0) months] and OS [31.5 (CI: 27.6–64.8) vs. 20.0 (CI: 14.0–36.7) months]. Concerning long-term survival after ten years, RT2 only lead to small improvements in OS [8% (CI: 1.4–45.3) vs. 0%]. RT2 improved survival most without (re)-resection [PFS: 17.5 (CI: 9.7–41.5) vs. 8.0 (CI: 6.6–12.2)/OS: 31.5 (CI: 27.6–NA) vs. 13.3 (CI: 8.1–20.1) months]. In the RT-naïve patients, CSI at recurrence improved their median PFS [25.0 (CI: 16.8–60.6) vs. 6.6 (CI: 1.5–NA) months] and OS [40.2 (CI: 18.7–NA) vs. 12.4 (CI: 4.4–NA) months]. Conclusions: RT2 could improve the median survival in a matched cohort but offered little benefit regarding long-term survival. In RT-naïve patients, CSI greatly improved their median and long-term survival.

## 1. Introduction

Radiotherapy (RT) is a key component of the standard treatment in medulloblastoma for patients of appropriate ages [1,2]. Irradiation is directed at the entire neuroaxis (cranio-spinal irradiation (CSI)) with an additional boost to the primary tumor site. Due to radiation dose restrictions and cumulative effects on neurocognitive abilities [3,4,5,6,7,8], re-irradiation (RT2) in cases of recurrence is therefore often a difficult decision for patients, their families and the physicians treating them. Some studies have shown that RT2 has the potential to improve disease control and survival [9,10,11,12,13].

As the nervous system of infants is especially susceptible to radiation-induced neurocognitive deficits, RT has been used more sparingly in patients under the ages of three to five years during initial treatment [14,15,16]. If recurrence occurs before delayed RT can be applied, these RT-naïve patients can be irradiated with the same approach that older patients normally receive during initial treatment. Previous analyses have shown that CSI after recurrence can still lead to long-term survival in a portion of patients [17,18,19].

Historically, medulloblastoma was divided into subgroups based on histology, which can be used as a predictor of survival. Now, medulloblastomas are subdivided into distinctive entities [20] based on genetic alterations, alternated pathways and methylation profiles [21]. These are now increasingly being used to predict survival and to individualize and stratify treatment [22,23,24,25] in an effort to maximize the survival chances while reducing the treatment burden in patients with less aggressive tumors [26].

In this analysis, we aim to (1) compare the survival of pre-irradiated patients treated with RT2 to those without further RT, using propensity score matching to control for possible biases in the non-randomized therapy selection, and (2) show the survival for patients without RT during initial treatment who at recurrence received either first-time irradiation (RT1) or were treated without it.

## 2. Materials and Methods

### 2.1. Cohort

Patient data from the German HIT-REZ 97 and HIT-REZ 2005 (NCT00749723) studies, as well as from the currently conducted HIT-REZ registry for patients with recurrent medulloblastoma, were gathered. The data included basic patient characteristics, tumor histology, treatment modalities and response during initial treatment and the treatment, response and outcome for all subsequent recurrences. Methylation-based molecular subgroups were included when molecular analysis was available either prospectively or retrospectively. All patient data regarding radiology, histology and molecular biology were reviewed by their respective centers during the patients’ participation in the study. Recommendations on radiotherapy were also given to all patients by central reference radiotherapists.

The eligibility criteria for inclusion within this analysis were an age of under 18 years at initial diagnosis, no M4 stage at first recurrence and sufficient data on treatment after initial diagnosis and first recurrence.

### 2.2. Treatment

Early postoperative MRI (up to 72 h postoperatively) was reviewed according to reference radiology to determine the extent of resection. If no residual tumor was found, resection was graded as a gross total resection (GTR). Near-total resection (NTR) was specified as contrast enhancement at the edge of the resection area with a reduction in tumor volume of at least 90%. Subtotal resection (STR) was defined as a reduction in tumor volume of less than 90%. Resections were also graded as STRs when single metastases were completely removed but other metastatic lesions remained. If no more than 10% of the tumor volume was resected, the surgery was deemed a biopsy. In cases enrolled before the standardized prospective molecular analyses, if tumor material from surgeries during initial treatment or recurrences were available, molecular data were analyzed retrospectively.

### 2.3. Statistical Analysis

All the statistical analyses were performed using R statistical programming (version 4.3.0; R Core Team, Vienna, Austria).

Progression-free survival (PFS) was defined as the time from diagnosis of first recurrence to either subsequent recurrence or death, while overall survival (OS) was defined as the time from diagnosis of first recurrence to death. If no applicable event occurred within a patient’s follow-up, right censoring at the time of last follow-up was used. Survival times and survival rates were estimated using the Kaplan–Meier method and presented as medians and their respective 95% confidence intervals (CIs). When lower or upper limits could not be calculated, these were designated as not applicable (NA). The results of the regression analyses are presented as hazard ratios (HRs) and their respective 95% CIs. All the survival analyses were performed using the survival package. Survival curves were generated using ggsurvplot() from the ggplot2 package.

Propensity score matching was used to balance the patient characteristics between the treated patients and untreated controls. Propensity scores (PS) and matching were performed using the MatchIt package. PS were estimated using a generalized linear model and matched 1:1 and without replacement (one control patient for every treated patient, no control patient was used more than one time) and using genetic matching with a population size of 1000 provided by the rgenoud package. To check for remaining data imbalances, standardized mean differences (SMDs) for all included covariates were calculated. SMDs under 0.1 were deemed acceptable. To find the best fitting model of genetic matching, 100 iterations of the matching were run, and the model producing the lowest cumulative levels of SMDs was chosen. Additional manual checking for imbalances was undertaken visually using bal.plot() and love.plot(), provided by the cobalt package.

### 2.4. Ethical Approval

All the procedures in this study involving human participants were in accordance with the ethical standards of the institutional and national committees. The trial was conducted in accordance with the 1964 Helsinki Declaration and its later amendments or comparable ethical standards. The institutional review boards or ethics committees of the leading centers at the universities of Bonn and Essen and all participating centers reviewed and approved the protocols of the HIT-REZ 97 and 2005 study as well as of the HIT-REZ registry. All parents/guardians and patients, where appropriate, gave their written informed consent for data collection and analysis.

## 3. Results

A total of 293 patients first diagnosed with recurrent medulloblastoma between 1973 and 2021 met the eligibility criteria (84.9% from a complete cohort of 345 patients) and were included within this analysis. A total of 116 (39.6%) were patients of the HIT-REZ ’97 study, 86 (29.4%) of the HIT-REZ 2005 study and 91 (31.1%) of the HIT-REZ registry. Table 1 shows an overview of the patient characteristics for the entire cohort and stratified by RT. The median follow-up was 18.7 months (IQR: 7.6–43.0) and 29.9 months (IQR: 12.1–68.7) for patients alive at last follow-up.

The median PFS from diagnosis of first recurrence for the entire cohort was 13.6 months (CI: 11.3–15.7), whereas the median OS was 26 months (CI: 20.6–31.7). The overall survival rates after 2 and 5 years were 51.7% (CI: 46–58.1) and 25.6% (CI: 20.5–32), respectively (Figure 1).

Methylation-based molecular subgroups were available for 107 patients (36.5%). Table 1 shows the rates at which the subgroups occurred in the complete cohort and by the radiotherapy applied. Of the patients with SHH medulloblastoma, 11 (68.8%) were under five years of age when they were first diagnosed. TP53 status was available for 8 patients with an SHH medulloblastoma (50%), which showed no alternation in 87.5%. Patients with Group 3 tumors showed the worst outcomes in both their median PFS [4.6 months (CI: 2.6–11)] and median OS [13.2 months (CI: 9–28.8)], while those with Group 4 tumors showed slightly better outcomes [PFS: 15.7 months (CI: 9.3–NA)/OS: NA months (CI: 15.7–NA)]. Patients with WNT medulloblastoma showed the highest five-year survival after recurrence at 52.4% (CI: 26.8–100%) (Figure 2).

### 3.1. Re-Irradiation

Propensity score matching was used to compare the patient group receiving RT2 to a control group of pre-irradiated patients without further RT at first recurrence but otherwise similar characteristics. The covariates for matching were sex, histology, disease status at the end of initial treatment, time to first recurrence, Chang stage, resection (GTR/NTR/STR vs. biopsy/no resection) and chemotherapy at first recurrence and whether radiotherapy was applied at subsequent relapses. A total of 42 patients treated with RT2 and 159 patients without RT had complete data on all covariates, and all patients with RT2 could be matched 1:1 to 42 control patients (Figure 3B). All confounders reached an SMD of <0.1 (Figure 3C). Due to the high amount of missing molecular data within the cohort, molecular subgroups could not be matched.

In the matched cohort, the median age at RT2 was 11.0 years (range: 4.5–29.6; IQR: 8.3–15.8), and the median time from the first day of initial radiotherapy to the first day of RT2 was 33.6 months (IQR: 22.5–51.2). Focal RT2 was applied at a median cumulative dose of 39.6 Gy (IQR: 30.0–47.1 Gy). Five patients received CSI as part of RT2 at a median cumulative dose to the entire neuroaxis of 42 Gy (IQR: 41.4–53.1).

Patients who received RT2 showed an improved median PFS of 21.0 months (CI: 15.7–28.7) compared to 12.0 months (CI: 8.1–21.0) in patients without RT2. An improvement in the median OS was also found for patients with RT2 at 31.5 months (CI: 27.6–64.8) compared to 20.0 months (CI: 14.0–36.7) (Figure 3). Advantages of RT2 were found in the survival rates after 2 years, both regarding PFS [39.1% (CI: 26.7–57.3) vs. 23.7% (CI: 13.6–41.4)] and OS [77.8% (CI: 66.0–91.7) vs. 39.9% (CI: 27.2–58.5)]. These advantages declined after 5 years both in the PFS rates [8.1% (CI: 2.4–27.8) vs. 7.9% (CI: 2.7–23.2)] and OS [(28.8% (CI: 16.4–50.5) vs. 16.9% (CI: 8.1–35.4)]. Both groups showed low OS rates after 10 years, with patients treated with RT2 standing at 8% (CI: 1.4–45.3) and patients without at 0%.

When the matched cohort was further stratified for the extent of resection at first recurrence (Figure 4), an advantage of re-irradiation compared to no RT2 in cases of no surgery or biopsy only (hereafter referred to as no resection) was found. Patients without resection and no RT2 (n = 25) had a median PFS of only 8.0 months (CI: 6.6–12.2), while this was 17.5 months (CI: 9.7–41.5) for patients treated with RT2 after no resection (n = 25). The same was found for the median OS, showing inferior results in patients with no resection and no RT2 [13.3 months (CI: 8.1–20.1)] compared to those treated with RT2 [31.5 months (CI: 27.6–NA)]. In contrast, no advantages in terms of survival were found for patients treated with RT2 after GTR, NTR or STR (n = 17 with RT2, n = 17 without RT2) concerning their median PFS [22.5 months with RT2 (CI: 17.5–58.1) vs. 21.1 months (CI: 16.5–34.3) without RT2] and their median OS [32.3 months with RT2 (CI: 26–NA) vs. 48 months (CI: 23.4–NA) without RT2].

Conversely, when looking only at patients who were treated with RT2, no improvements in survival were found when they also received GTR, NTR or STR at recurrence (n = 16) compared to no additional surgery (n = 23). The median PFS in patients with RT2 and resection at recurrence was 18.4 months (CI: 16.8–NA) compared to RT2 without resection at 25 months (CI: 15.5–NA). Similarly, the median OS for RT2 and resection was 21.8 months (CI: 18.4–NA) compared to RT2 without resection at 56.7 months (CI: 26.7–NA).

Performing a multivariate Cox regression for the application of RT2 and all the matched covariates showed survival advantages in both the PFS and OS for RT2 [HR: PFS = 0.55 (CI: 0.33–0.92), OS = 0.41 (CI: 0.23–0.73)]. For the PFS regression, the application of chemotherapy at recurrence [HR: 0.41 (CI: 0.14–1.19)] and a time to recurrence after initial diagnosis of over 18 months [HR: 0.07 (CI: 0.03–0.16)] also lowered the patients’ HRs. Regarding OS, a time to recurrence of over 18 months showed considerably improved HRs [0.04 (CI: 0.01–0.10)].

The five patients who received CSI as part of RT2 showed, in comparison to patients with focal RT2 only, no improved median PFS [17.5 months (CI: 4.6–NA) vs. 22.8 months (CI: 12.3–50.4)] or median OS [28.8 months (CI: 9–NA) vs. 30 months (CI: 25.6–NA)].

### 3.2. First Radiotherapy at First Recurrence

Patients treated without RT during first-line treatment had a median age of 2.7 years (range: 0.2–12.7, IQR: 1.7–3.5 days) at first diagnosis. As seen in Table 1, patients who received RT neither at initial diagnosis nor at first recurrence had a median age of under 3 years when recurring, while patients with RT1 had a higher median age of 4.6 years (IQR: 3.4–6.3). All the patients who received RT1 received irradiation of the entire neuroaxis at first recurrence, with a median dose of 35.2 Gy (IQR: 24–35.2) and a focal boost to 55.0 Gy (IQR: 54.0–55.0). All patients without RT during initial treatment received surgery and adjuvant chemotherapy.

Due to the small number of patients without RT1 within this cohort of radiotherapy-naïve patients, no propensity score matching could be performed, limiting the comparability between the treatment groups. When comparing the groups, patients treated with RT1 showed an improved survival (Figure 5) both regarding their median PFS [25.0 months (CI: 16.8–60.6) vs. 6.6 months (CI: 1.5–NA)] and median OS [40.2 months (CI: 18.7–NA) vs. 12.4 months (CI: 4.4–NA)]. RT1 showed beneficial results both in terms of local recurrences (median PFS: 25.2 months [CI: 16.8–NA)] vs. 6.6 months [CI: 1.5–NA]; median OS: 60.5 months [CI: 18.7–NA] vs. 28.5 months [CI: 12.1–NA]) and in terms of metastatic recurrences (median PFS: 20.7 months [CI: 15.5–NA)] vs. 5.5 months [CI: 1.5–NA]; median OS: 40.2 months [CI: 16.4–NA] vs. 10.0 months [CI: 1.5–NA]).

The positive impact of RT1 on survival was most noticeable in the patients with no previous resection at first recurrence. When RT1 was applied within this group, the median PFS was 25.0 months (CI: 15.5–NA) compared to only 3.0 months (CI: 1.5–NA) without RT. The median OS was likewise improved at 56.7 months (CI: 26.7–NA) compared to 8.3 months (CI: 1.5–NA), respectively. However, in contrast to patients treated with RT2, patients with RT1 also showed improved survival compared to patients without RT in cases with previous surgery. Here, the median PFS with RT1 was 18.4 months (CI: 16.8–NA) compared to 6.7 months (CI: 3.4–NA) without RT1, and the median OS was 21.8 months (CI: 18.4–NA with) compared to 14.9 months (CI: 7.7–NA without RT1).

Importantly, the patients with RT1 showed a 10-year OS rate of 33.1% (CI: 19.3–56.7), while no patients without RT1 were alive at this time point.

Using multivariate Cox regression of the same variables as for the RT2 cohort (excluding surgery and chemotherapy at initial diagnosis, which all received), we found RT1 to be associated with a vast improvement in PFS [HR: 0.10 (CI: 0.04–0.29)] and in OS [HR: 0.10 (CI: 0.04–0.29)]. Further factors associated with an improvement in PFS were chemotherapy at first recurrence [HR: 0.13 (CI: 0.04–0.39)] and a time to first recurrence of over 18 months [HR: 0.15 (CI: 0.05–0.52)]. Regarding OS, improvements were found for the same factors, at HRs of 0.08 (CI: 0.02–0.27) and 0.12 (CI: 0.03–0.42) for chemotherapy and time to recurrence, respectively.

In neither the RT2 nor the RT1 group were any radiation-induced severe adverse effects of CTCAE grade 4 or higher reported. No data were acquired on the neurocognitive outcomes of the patients after recurrence.

## 4. Discussion

Our analysis evaluated the survival outcomes for patients with recurrent medulloblastoma, stratified by previous radiotherapy during initial therapy. Regarding the pre-irradiated patients, using propensity score matching to reduce heterogeneity and possible biases in the choice of whether radiotherapy was applied again, we found advantages of treating patients ineligible for resection with RT2. No advantage of RT2 in addition to surgery with at least subtotal resection was found. It also has to be noted that the patient cohort who received RT2 showed only eight percent survivors past approximately 10 years after the diagnosis of first recurrence.

Previous studies on RT2 in patients with recurrent medulloblastoma have mostly been conducted with small case numbers and have shown that long-term survival and disease control may be achievable. Bakst et al. reported a 65% OS after five years (n = 13) [27], while Wetmore et al. reported a 45% OS after ten years for standard-risk patients treated with RT2 (n = 38, 14 patients with RT2) [11]. This stands in contrast to the 10-year-OS of 8% found within our analysis, which is closer to the long-term survival rates reported by other groups [9,13]. However, after 2 years, our analysis showed better OS results (82.4%) than 64%, reported by Gupta et al. [28], and only 25% after three years, reported by Milker-Zabel [9].

We found no benefit of CSI as part of RT2, which differs from the results of Baroni et al., who reported a 50% OS after three years for RT2-CSI but of 0% for focal RT2, while also showing the large impact on cognitive development brought on by multiple RTs of the entire neuroaxis [3]. However, the number of patients receiving CSI in our cohort was very limited, and further studies are needed to compare both focal RT2 as well as RT2 with CSI to no re-irradiation.

Superior outcomes after surgery and RT2 were previously also reported by Bakst et al. in a small study of 13 patients, who found improved rates of disease-free survival when RT2 was applied after GTR, with survival mostly being determined by the presence of residual tumors at the start of RT2 [27]. In a more recent analysis by Gupta et al., however, no significant impact of re-resection on the survival of patients treated with RT2 was found [28]. Other studies have also shown benefits of treating patients with re-resection when possible [29]. In our cohort, the patients showed no additional survival benefit when they were treated with both RT2 and resection compared to only RT2.

In the other cohort of patients who had not previously received radiotherapy, a clear advantage of the use of radiotherapy was found, with a median OS of 40.2 months. These results are in line with the well-established role of radiotherapy during initial treatment [2,30]. When postponement of RT in young patients leads to progression or relapse, RT1 can be initiated at recurrence without dose restrictions. In a cohort of 60 patients without upfront CSI, Hill et al. found similar results to those we have presented here [17]. Multiple other studies have also found long-term survival with CSI and focal boost in RT-naïve patients [31,32,33]. Our results also confirm preliminary results from a previous report on a subsection of patients from the HIT-REZ 97 and 2005 studies [18].

Patients with recurrent SHH medulloblastoma were comparatively rare in our cohort, making up 15% of the 107 patients with a known molecular subgroup, and showed a better survival outcome than Groups 3 and 4. This stands in contrast to previously published data which reported SHH medulloblastoma to be associated with worse outcomes compared to Group 4 medulloblastoma and showed higher rates of this subgroup at recurrence [17,34,35]. This may be due to an enrichment of patients under 5 years at initial diagnosis, who made up 68.8% of the patients with recurrent SHH medulloblastoma in our cohort and who were previously indicated to represent different SHH subtypes with different underlying gene alterations [36,37,38]. Accordingly, patients with SHH medulloblastoma and known TP53 status showed wild-type TP53 in most cases, which has also shown to be associated with a better outcome in SHH medulloblastoma [39,40]. Both the dismal survival after relapse with Group 3 medulloblastoma [17,34] and the good survival for WNT medulloblastoma [39] found within our cohort are in line with previously published results.

The presented study is limited by the lack of molecular data, as many patients from older studies were included and the relatively low rates of re-resection at recurrence hindered broad retrospective analyses of the tumor material. The propensity score matching used to limit biases between cohorts could therefore not be adjusted to molecular groups, which have been shown to have a significant impact on survival. Additionally, even though the matching controlled for the chemotherapy received, no matching was possible for the chemotherapy regimens used. Therefore, bias due to different intensities of systemic and intraventricular chemotherapy cannot be ruled out, as previous works have shown possible survival benefits when local therapy was not possible [41,42,43,44]. Furthermore, while matching can reduce the differences between treated patients and controls, less quantifiable variabilities in the choices for or against RT2 might still have accounted for hidden biases, which we could not correct for.

## 5. Conclusions

We show that second RT at recurrence improves survival compared to a similarly matched cohort of control patients without second RT in cases in which previous re-resection was not possible. However, no additional benefit could be found in our cohort for its use after tumor-debulking surgery. Therefore, RT2 might be an effective alternative to surgery at recurrence. In consideration of the additive detrimental effects of multiple rounds of radiation on the neurocognitive development of young patients, cautious usage seems advisable. In patients too young to receive RT at first-line treatment, our data indicate that primary RT at recurrence is associated with markedly improved survival. Due to the non-randomized, retrospective order in which our analysis was conducted, prospective trials or retrospective multinational meta-analyses with comprehensive molecular analyses would be needed to offer more reliable data and inform future treatment decisions.

## Figures and Tables

**Figure 1 cancers-16-01955-f001:**
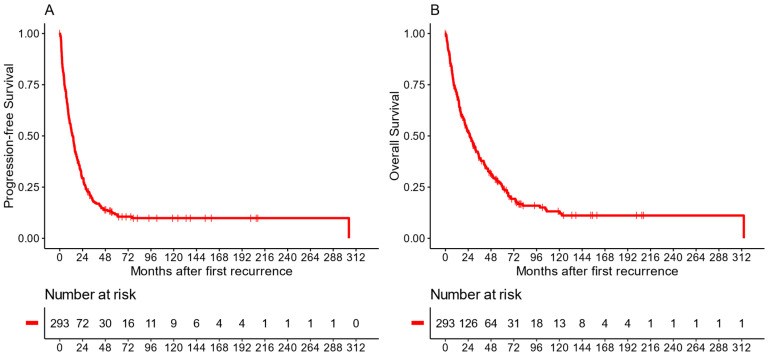
Kaplan–Meier plots of the entire cohort, with panel (**A**) showing PFS and panel (**B**) showing OS.

**Figure 2 cancers-16-01955-f002:**
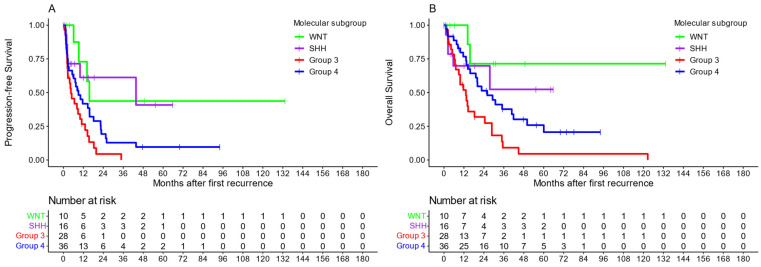
Kaplan–Meier plots of all patients with known subgroups, stratified by molecular subgroup. Panel (**A**) shows PFS; panel (**B**) shows OS.

**Figure 3 cancers-16-01955-f003:**
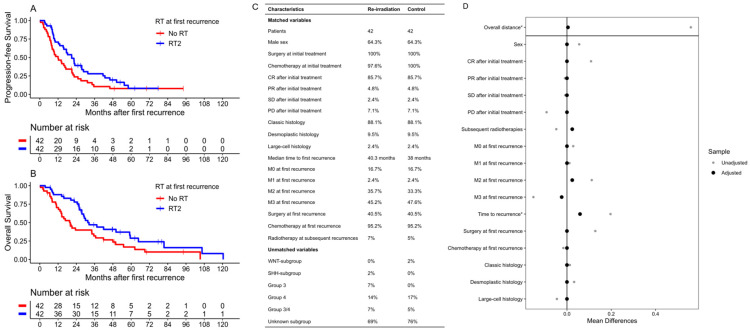
Kaplan–Meier plots ((**A**) for PFS, (**B**) for OS) for the matched cohort of patients with re-irradiation (RT2) compared to control patients without re-irradiation (No RT); and patient characteristics ((**C**) as a table comparing ratios and medians, (**D**) showing love plots of all matched characteristics and overall distance in propensity scores) compared between treated and control subjects. The symbol * designates continuous variables, for which the standardized mean difference is given.

**Figure 4 cancers-16-01955-f004:**
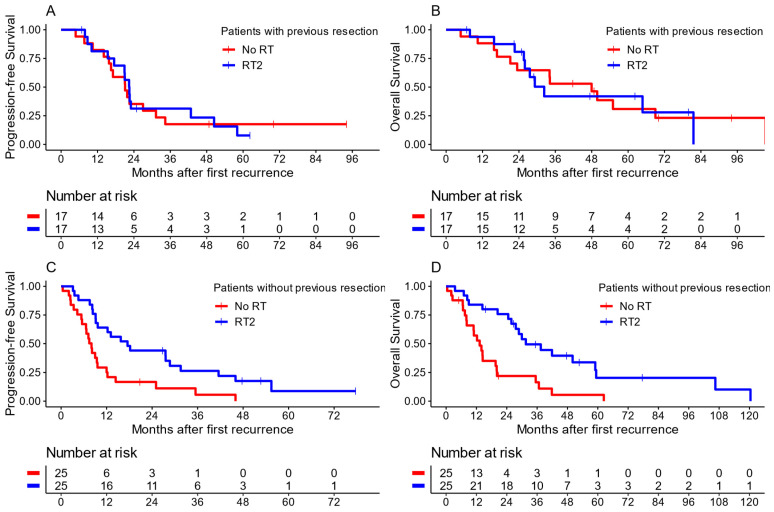
Kaplan–Meier plots for the matched cohort, comparing re-irradiation (RT2) to no radiotherapy at first recurrence (No RT). Panels (**A**) (PFS) and (**B**) (OS) show patients with at least subtotal resection at first recurrence, while panels (**C**) (PFS) and (**D**) (OS) show patients without resection at first recurrence.

**Figure 5 cancers-16-01955-f005:**
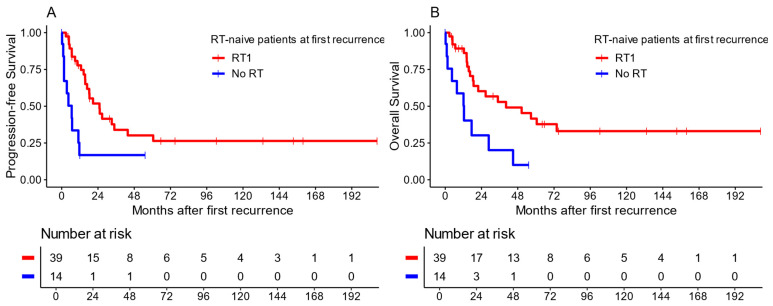
Kaplan–Meier plots comparing survival for patients without previous radiotherapy, stratified by radiotherapy at recurrence. Panel (**A**) shows PFS; panel (**B**) shows OS.

**Table 1 cancers-16-01955-t001:** Patient characteristics for the entire cohort as well as stratified by radiotherapy.

Characteristic	Overall, n = 293 ^1^	First Irradiation, n = 39 ^1^	Re-Irradiation, n = 56 ^1^	No Re-Irradiation, n = 184 ^1^	No Radiation, n = 14 ^1^
Sex					
male	206 (70%)	31 (79%)	40 (71%)	128 (70%)	7 (50%)
female	87 (30%)	8 (21%)	16 (29%)	56 (30%)	7 (50%)
Age at first recurrence; years (IQR)	9.4 (6.2, 13.0)	4.6 (3.4, 6.3)	11.2 (8.4, 14.7)	11.0 (7.8, 14.2)	2.7 (2.1, 3.5)
Time to first recurrence from initial diagnosis; months (IQR)	21.6 (15.1, 42.1)	16.2 (13.7, 20.5)	32.7 (23.2, 47.7)	22.2 (15.6, 42.9)	14.8 (8.8, 15.5)
Molecular subgroup					
WNT	10 (9.3%)	1 (5.9%)	0 (0%)	9 (13%)	0 (0%)
SHH	16 (15%)	7 (41%)	1 (6.7%)	4 (5.8%)	4 (67%)
Group 3	28 (26%)	7 (41%)	3 (20%)	16 (23%)	2 (33%)
Group 4	36 (34%)	1 (5.9%)	7 (47%)	28 (41%)	0 (0%)
Group 3/4	17 (16%)	1 (5.9%)	4 (27%)	12 (17%)	0 (0%)
Resection during initial treatment	280 (96%)	39 (100%)	53 (95%)	174 (95%)	14 (100%)
GTR	167 (60%)	27 (69%)	34 (64%)	98 (56%)	8 (57%)
NTR	74 (26%)	8 (21%)	12 (23%)	50 (29%)	4 (29%)
STR	35 (13%)	4 (10%)	7 (13%)	23 (13%)	1 (7.1%)
Biopsy	4 (1.4%)	0 (0%)	0 (0%)	3 (1.7%)	1 (7.1%)
Chemotherapy during initial treatment	288 (98%)	37 (95%)	54 (96%)	183 (99%)	14 (100%)
Chang stage at first recurrence					
M0	51 (17%)	12 (31%)	9 (16%)	26 (14%)	4 (29%)
M1	6 (2.0%)	1 (2.6%)	2 (3.6%)	2 (1.1%)	1 (7.1%)
M2	83 (28%)	11 (28%)	21 (38%)	46 (25%)	5 (36%)
M3	153 (52%)	15 (38%)	24 (43%)	110 (60%)	4 (29%)
M4	0 (0%)	0 (0%)	0 (0%)	0 (0%)	0 (0%)
Resection at first recurrence	117 (40%)	16 (41%)	33 (59%)	61 (33%)	5 (36%)
GTR	54 (46%)	7 (39%)	12 (36%)	33 (54%)	2 (40%)
NTR	18 (15%)	5 (28%)	5 (15%)	7 (11%)	1 (20%)
STR	16 (14%)	4 (22%)	3 (9.1%)	7 (11%)	2 (40%)
Biopsy	29 (25%)	2 (11%)	13 (39%)	14 (23%)	0 (0%)
Chemotherapy at first recurrence	267 (91%)	33 (85%)	53 (95%)	170 (92%)	11 (79%)
CSI at first recurrence	48 (16%)	36 (92%)	12 (21%)	0 (0%)	0 (0%)

^1^ n (%); median (IQR), IQR = interquartile range; GTR = gross total resection; NTR = near-total resection, STR = subtotal resection; CSI = craniospinal irradiation.

## Data Availability

The anonymized data presented in this study as well as all the software code are available on request from the corresponding author.

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
