# Peer review of "Radiotherapy for Recurrent Medulloblastoma in Children and Adolescents: Survival after Re-Irradiation and First-Time Irradiation"

_cancers, 2024, doi:10.3390/cancers16111955_

Round 1

Reviewer 1 Report

Comments and Suggestions for Authors

Referee's report

Brief Summary

Treatment options for relapsed medulloblastoma is a challenging topic for the pediatric neuro-oncologist. The authors provide a well written and robust contribution to this discussion, which fits the journal’s scope. The use of a matched cohort strengthens the non-randomized results so that they allow a more precise estimation of treatment effects. The retrospective molecular analyses, even if only available in a minority of patients and not impactful on results, are especially significant in the contemporary landscape of medulloblastoma. The study highlight the role of re-irradiation and first radiation at time of relapse of medulloblastoma in children.

I evaluate the manuscript suitable of publication in your prestigious Journal with only some minor revisions specified below point-by-points.

Concept and specific comments

1.       At line 101 of Material and Methods: the Authors defined STR as “a reduction in tumor volume of less than 90%. Resections were also graded as STR when single metastases were completely removed, but other metastatic lesions remained." The thus defined STR group represents in my opinion a too heterogeneous subgroup, potentially including metastatic disease at relapse and an higher chance of gross residual disease at re-irradiation.  Even if it is a small subgroup (14 patients) it could impact on results and conclusions made by the Authors that included STR. If the numbers allowed it, a separate survival analysis for STR+RT2 patients might still show a benefit like the one observed for RT2 without resection patients. Alternatively this bias could  be addressed in the discussion more clearly.

2.       At line 102 of the Material and Methods section: it is stated that recommendation on radiotherapy were given to all patient by central reference radiotherapists, I wonder the Authors could specify the criteria, besides age at relapse and time to front line RT, generally applied to give recommendation to  RT2 and extent/CSI.

3.       Line 158 in Table 1: In order to help readers, it could be useful to identify more clearly the total number of patients who underwent surgery, adding a line with surgery yes/no as part of first line treatment and at recurrence and not only extent of resection.

4.       At line 310 in the discussion section: I recommend the Author to rephrase their comment of the paper by Bakst et al. [27]. It should be noted that the cited paper only included 13 patients and only 5 of them with GTR at recurrence (7 patients had no surgery at all at recurrence). The difference in outcome was determined by the presence of gross residual disease at time of re-irradiation (mainly, but not exclusively determined by extent of surgery), and as stated by the authors of the cited paper, the exact contribution of each modality is difficult to determine due to study limitations. Due to this reasons a minor emphasis regarding outcome of RT2 in GTR patients, is warranted.

5.       At line 351 In the conclusion section: I suggest to mitigated the sentence “Therefore, RT2 can be considered as an alternative to surgery at recurrence”.  Data shown, in my opinion, underline more clearly that in a multimodal and individualised approach to medulloblastoma recurrence, RT2 might be an effective alternative to surgery.

6.       Patients who received CSI at recurrence were mostly RT-naïve patients; only 12 patients (5/42 in the matched cohort) received CSI re-irradiation. At line 308 the authors correctly recognise this limitation in the discussion. Could the Authors speculate about a separate survival analysis including 3 groups: No-RT2, RT2-Focal and RT2-CSI maybe as future prospective. Moreover in the discussion it will be informative, if available, data on radiotherapy technique used at recurrence, maybe as supplementary materials (e.g.: average cumulative dose, photons, protons, fractioning, etc.).

7.       An interesting addition to the data presented by the Authors, if available and as supplementary materials, some details about chemotherapy done at relapse across protocols of treatment (e.g. patients received high-dose chemotherapy with autologous stem cell rescue).

In the Discussion section: Considering that 91% of patients received some chemotherapy at relapse, could be interesting if the Authors could speculate about chemotherapy role in the relapse setting, even if I understand this is not the main focus of the study.

Author Response

At line 101 of Material and Methods: the Authors defined STR as “a reduction in tumor volume of less than 90%. Resections were also graded as STR when single metastases were completely removed, but other metastatic lesions remained." The thus defined STR group represents in my opinion a too heterogeneous subgroup, potentially including metastatic disease at relapse and an higher chance of gross residual disease at re-irradiation.  Even if it is a small subgroup (14 patients) it could impact on results and conclusions made by the Authors that included STR. If the numbers allowed it, a separate survival analysis for STR+RT2 patients might still show a benefit like the one observed for RT2 without resection patients. Alternatively this bias could  be addressed in the discussion more clearly.

Answer: We investigated this to see whether patients with resections assessed as STR but with multiple metastases could have biased the results. The RT2 group involved one patient with STR, who had a local first recurrence which was sub-totally resected. Due to the matching the control group only involved one patient with STR as well. In this case the patient had two distinct spinal metastases which were both sub-totally resected.  Therefore, it is correct that this may have led to a small bias, though the metastatic state should have mostly been mitigated within the matching process ahead of survival analyses. Thus, because of only one patient having RT2 + STR, a further analysis of survival in this patient group was unfortunately not possible.

At line 102 of the Material and Methods section: it is stated that recommendation on radiotherapy were given to all patient by central reference radiotherapists, I wonder the Authors could specify the criteria, besides age at relapse and time to front line RT, generally applied to give recommendation to RT2 and extent/CSI.

Answer: It is generally difficult to provide specific criteria with which the suggestions were made by the central reference radiotherapists, as these changed over time and were informed by the current standards of therapy within the literature and were not given defined by the respective study protocols themselves.

Line 158 in Table 1: In order to help readers, it could be useful to identify more clearly the total number of patients who underwent surgery, adding a line with surgery yes/no as part of first line treatment and at recurrence and not only extent of resection.

Answer: The total amount of patients having undergone surgery at both initial therapy and recurrence and their percentages have now been provided within table 1.

At line 310 in the discussion section: I recommend the Author to rephrase their comment of the paper by Bakst et al. [27]. It should be noted that the cited paper only included 13 patients and only 5 of them with GTR at recurrence (7 patients had no surgery at all at recurrence). The difference in outcome was determined by the presence of gross residual disease at time of re-irradiation (mainly, but not exclusively determined by extent of surgery), and as stated by the authors of the cited paper, the exact contribution of each modality is difficult to determine due to study limitations. Due to this reasons a minor emphasis regarding outcome of RT2 in GTR patients, is warranted.

Answer: Both the small case number as well as survival being mostly determined by the amount of residual disease at RT2 have now been discussed within the paragraph of the discussion.

At line 351 In the conclusion section: I suggest to mitigated the sentence “Therefore, RT2 can be considered as an alternative to surgery at recurrence”.  Data shown, in my opinion, underline more clearly that in a multimodal and individualised approach to medulloblastoma recurrence, RT2 might be an effective alternative to surgery.

Answer: The sentenced has been mitigated and instead the suggested phrasing of and effective alternative to surgery has been used.

Patients who received CSI at recurrence were mostly RT-naïve patients; only 12 patients (5/42 in the matched cohort) received CSI re-irradiation. At line 308 the authors correctly recognise this limitation in the discussion. Could the Authors speculate about a separate survival analysis including 3 groups: No-RT2, RT2-Focal and RT2-CSI maybe as future prospective. Moreover in the discussion it will be informative, if available, data on radiotherapy technique used at recurrence, maybe as supplementary materials (e.g.: average cumulative dose, photons, protons, fractioning, etc.).

Answer: Data on the cumulative dose has now been provided within the results section on the patients receiving RT2, additionally to those concerning RT1 and CSI already included within the manuscript. Data concerning photon/proton therapy as well as fractioning was unfortunately not generally gathered. We have data only the fractioning of eleven patients, of which only one patient received a hypofractionated irradiation. A further sentence discussing RT2-focal and RT2-CSI compared to no re-irradiation has been added within the discussion.

An interesting addition to the data presented by the Authors, if available and as supplementary materials, some details about chemotherapy done at relapse across protocols of treatment (e.g. patients received high-dose chemotherapy with autologous stem cell rescue).

Answer: Unfortunately, due to chemotherapy not being the main focus of our manuscript, this data is still currently in the process of being worked up for the entire cohort presented in this manuscript. Therefore, no broader statements can be made about the protocols and specific drugs used at the current time. This is however an active work in progress which we plan to further elaborate on when we have pooled all the data. Most patients included from the HIT-REZ 97 and HIT-REZ 2005 studies would have received their respective protocols, which were carboplatin + etoposide for the 97 study, and two arms of either carboplatin + etoposide or temozolomide in the 2005 study. Data on chemotherapy within both studies have been previously published (Bode et al. 2014 in Journal of Neuro-Oncology and Gaab et al. 2022 in Cancers).

In the Discussion section: Considering that 91% of patients received some chemotherapy at relapse, could be interesting if the Authors could speculate about chemotherapy role in the relapse setting, even if I understand this is not the main focus of the study.

Answer: We have added the use of chemotherapy, especially as it pertains to possibly more intensive chemotherapy in the control group without further radiotherapy, as a possible bias within our results. We now discuss this in the last paragraph of the discussion.

Reviewer 2 Report

Comments and Suggestions for Authors

Overall, the paper is well organized and presented, with clear conclusions. Some minor comments:

1. Number should be spelled out when a sentence starts with it. Therefore, the sentence "293 patients first diagnosed with recurrent medulloblastoma between 1973 and 2021 met eligibility criteria (84.9% from complete cohort of 345 patients) and were included within this analysis." and "116 (39.6%) were patients of the HIT‑REZ ’97 study, 86 (29.4%) of the HIT‑REZ 2005 study and 91 (31.1%) of the HIT‑REZ registry." between lines 151 and 154 of page 4 should be modified accordingly. Another case is "42 patients treated with RT2 and 159 patients without RT had complete data on all covariates, and all patients with RT2 could be matched 1:1 to 42 control patients (Figure 3b)" between lines 185 and 187 of page 6;

2. For Figure 2, the sentence "Kaplan-Meier plots of all patients with known subgroups, stratified by molecular sub-group." within the title between lines 177 and 178 of page 5 should not be bolded;

3. For Figure 5, although it is obvious for the left and right subfigures to represent panel A and B respectively, labels A and B are missing. Please add A and B in the subplot properly;

4. For Table 1, the layout should modify properly. In the first column "Characteristics", as the items "Sex", "Molecular subgroup", "Extent of resection during initial treatment", "Chang stage at first recurrence", and "Extent of resection at first recurrence" contain subitems, they should be put in a single sub-column under the "Characteristics" to present them in a clear way. To make it clear, the single column under "Characteristics" should divided into two sub-columns, in which the left sub-column contains the items "Sex", "Molecular subgroup", "Extent of resection during initial treatment", "Chang stage at first recurrence", and "Extent of resection at first recurrence", and their corresponding subitems placed in the right sub-column accordingly;

5. For this sentence "Accordingly, patients with SHH‑medulloblastoma and known TP53‑status showed wildtype TP53 in most cases, which were also shown to be associated with better outcome in SHH‑medulloblastoma.[39,40]" between lines 334 and 336 of page 10, "[39,40]" should be placed in front of the period ".".

Comments on the Quality of English Language

Overall, the paper is well organized and presented, with clear conclusions. Some minor comments:

1. Number should be spelled out when a sentence starts with it. Therefore, the sentence "293 patients first diagnosed with recurrent medulloblastoma between 1973 and 2021 met eligibility criteria (84.9% from complete cohort of 345 patients) and were included within this analysis." and "116 (39.6%) were patients of the HIT‑REZ ’97 study, 86 (29.4%) of the HIT‑REZ 2005 study and 91 (31.1%) of the HIT‑REZ registry." between lines 151 and 154 of page 4 should be modified accordingly. Another case is "42 patients treated with RT2 and 159 patients without RT had complete data on all covariates, and all patients with RT2 could be matched 1:1 to 42 control patients (Figure 3b)" between lines 185 and 187 of page 6;

2. For Figure 2, the sentence "Kaplan-Meier plots of all patients with known subgroups, stratified by molecular sub-group." within the title between lines 177 and 178 of page 5 should not be bolded;

3. For Figure 5, although it is obvious for the left and right subfigures to represent panel A and B respectively, labels A and B are missing. Please add A and B in the subplot properly;

4. For Table 1, the layout should modify properly. In the first column "Characteristics", as the items "Sex", "Molecular subgroup", "Extent of resection during initial treatment", "Chang stage at first recurrence", and "Extent of resection at first recurrence" contain subitems, they should be put in a single sub-column under the "Characteristics" to present them in a clear way. To make it clear, the single column under "Characteristics" should divided into two sub-columns, in which the left sub-column contains the items "Sex", "Molecular subgroup", "Extent of resection during initial treatment", "Chang stage at first recurrence", and "Extent of resection at first recurrence", and their corresponding subitems placed in the right sub-column accordingly;

5. For this sentence "Accordingly, patients with SHH‑medulloblastoma and known TP53‑status showed wildtype TP53 in most cases, which were also shown to be associated with better outcome in SHH‑medulloblastoma.[39,40]" between lines 334 and 336 of page 10, "[39,40]" should be placed in front of the period ".".

Author Response

Number should be spelled out when a sentence starts with it. Therefore, the sentence "293 patients first diagnosed with recurrent medulloblastoma between 1973 and 2021 met eligibility criteria (84.9% from complete cohort of 345 patients) and were included within this analysis." and "116 (39.6%) were patients of the HIT‑REZ ’97 study, 86 (29.4%) of the HIT‑REZ 2005 study and 91 (31.1%) of the HIT‑REZ registry." between lines 151 and 154 of page 4 should be modified accordingly. Another case is "42 patients treated with RT2 and 159 patients without RT had complete data on all covariates, and all patients with RT2 could be matched 1:1 to 42 control patients (Figure 3b)" between lines 185 and 187 of page 6;

Answer: This has been modified within the revised manuscript.

For Figure 2, the sentence "Kaplan-Meier plots of all patients with known subgroups, stratified by molecular sub-group." within the title between lines 177 and 178 of page 5 should not be bolded;

Answer: This has been modified within the revised manuscript.

For Figure 5, although it is obvious for the left and right subfigures to represent panel A and B respectively, labels A and B are missing. Please add A and B in the subplot properly;

Answer: Figure 5 has been corrected accordingly.

For Table 1, the layout should modify properly. In the first column "Characteristics", as the items "Sex", "Molecular subgroup", "Extent of resection during initial treatment", "Chang stage at first recurrence", and "Extent of resection at first recurrence" contain subitems, they should be put in a single sub-column under the "Characteristics" to present them in a clear way. To make it clear, the single column under "Characteristics" should divided into two sub-columns, in which the left sub-column contains the items "Sex", "Molecular subgroup", "Extent of resection during initial treatment", "Chang stage at first recurrence", and "Extent of resection at first recurrence", and their corresponding subitems placed in the right sub-column accordingly;

Answer: We have made according changes to improve the readability of table 1. The main characteristics are now shown left-bound, while the sub-columns are indented.

For this sentence "Accordingly, patients with SHH‑medulloblastoma and known TP53‑status showed wildtype TP53 in most cases, which were also shown to be associated with better outcome in SHH‑medulloblastoma.[39,40]" between lines 334 and 336 of page 10, "[39,40]" should be placed in front of the period ".".

Answer: This has been modified within the revised manuscript.